# Parameter tuning differentiates granule cell subtypes enriching transmission properties at the cerebellum input stage

Stefano Masoli [1], Marialuisa Tognolina [1], Umberto Laforenza[2], Francesco Moccia[3] & Egidio D'Angelo [1,4✉]

The cerebellar granule cells (GrCs) are classically described as a homogeneous neuronal population discharging regularly without adaptation. We show that GrCs in fact generate diverse response patterns to current injection and synaptic activation, ranging from adaptation to acceleration of firing. Adaptation was predicted by parameter optimization in detailed computational models based on available knowledge on GrC ionic channels. The models also predicted that acceleration required additional mechanisms. We found that yet unrecognized TRPM4 currents specifically accounted for firing acceleration and that adapting GrCs outperformed accelerating GrCs in transmitting high-frequency mossy fiber (MF) bursts over a background discharge. This implied that GrC subtypes identified by their electroresponsiveness corresponded to specific neurotransmitter release probability values. Simulations showed that fine-tuning of pre- and post-synaptic parameters generated effective MF-GrC transmission channels, which could enrich the processing of input spike patterns and enhance spatio-temporal recoding at the cerebellar input stage.

---

[1] Department of Brain and Behavioral Sciences, University of Pavia, Via Forlanini 6, 27100 Pavia, Italy. [2] Department of Molecular Medicine, University of Pavia, Via Forlanini 6, 27100 Pavia, Italy. [3] Department of Biology and Biotechnology, University of Pavia, Via Forlanini 6, 27100 Pavia, Italy. [4] Brain Connectivity Center, IRCCS Mondino Foundation, Via Mondino 2, 27100 Pavia, Italy. These authors contributed equally: Stefano Masoli, Marialuisa Tognolina. ✉email: dangelo@unipv.it

The intrinsic variability in the ionic currents, the neuron's morphology, and the neurotransmitter release dynamics are thought to be crucial for generating the richness of circuit properties and information-carrying capacity of brain microcircuits[1–3]. A puzzling case is presented by cerebellar granule cells (GrC), the most numerous neurons of the brain[4]. GrCs are located at the input stage of cerebellum, where they are thought to perform the fundamental operations of combinatorial expansion and spatio-temporal recoding predicted by the motor learning theory[5,6]. While these operations would greatly benefit of a rich repertoire of signal transformation properties, GrCs appeared very homogeneous in size and shape since their original description[7,8]. Later on, electrophysiological recordings from the cerebellar vermis of rodents reported a stereotyped firing pattern with spikes organized in regular discharges with little or no adaptation[9–11]. The experimental identification of a basic set of eight ionic mechanisms[12] allowed to model GrC firing in great detail[13–16], leading to a "canonical" description of GrCs but the question about the potential differentiation of firing properties remained open.

In a recent work[17], automatic optimization procedures were applied to GrC models yielding a family of solutions with maximum ionic conductance values falling within the range of physiological variability. During short current injections (500–800 ms, as used in previous experiments), all GrC models conformed to the canonical firing pattern but, unexpectedly, prolonged current injections (2 s) revealed a rich repertoire of adaptation properties. Here, this prediction was tested experimentally by whole-cell recordings, which indeed revealed various degrees of firing adaptation and, in addition, also firing acceleration in some GrCs. This observation not just supported that conductance tuning would allow the emergence of a richness of electro-responsive properties, but also implied the presence of yet unrecognized ionic mechanism causing firing acceleration.

Among the possible mechanisms causing firing acceleration there are transient receptor potential (TRP) channels[18–26], which can generate delayed depolarizing currents following intense firing and consequent activation of intracellular cascades. Indeed, we have been able to measure TRP Melastatin 4 (TRPM4) currents in accelerating GrCs. These results suggest that GrC complexity in the cerebellar vermis is higher than previously thought, raising the number of ionic conductances required to determine the firing pattern. It was already known that GrC of vestibulo-cerebellum are specialized to slow-down firing modulation based on the expression of low-threshold $Ca^{2+}$ channels[27]. Therefore, despite their morphological homogeneity, GrCs have differentiated conductance tuning and ionic channel expression, which could be further modified by fine variants in dendritic/axonal organization[28].

On a different scale, mossy fibers (MFs) convey to GrCs combinations of burst and protracted frequency-modulated discharges[29–31] lasting up to several seconds (e.g., see refs. [32,33]) that are dynamically transmitted at the MF-GrC synapses exploiting short-term plasticity mechanisms fine-tuned by vesicle release probability ($p$)[34–36]. Here, we observed that adapting GrCs exploit low-$p$ synapses attaining a much higher signal-to-noise ratio ($S/N$) than accelerating GrCs.

These results show that, following Getting's (1989) predictions[1], parameter variability at different scales supports the emergence of a richness of properties of potential physiological relevance in cerebellar GrCs. Tuning of adaption/acceleration and short-term plasticity generated a rich repertoire of filtering properties[37,38], which could substantially contribute to spatio-temporal recoding of synaptic input patterns at the cerebellum input stage[5].

## Results

### Different electroresponsive properties in cerebellar GrCs. In order to accurately assess the GrC firing properties, whole-cell recordings were carried out in current-clamp configuration while delivering 2-s current steps at different intensities from the holding potential of –65 mV. This protocol differs from those used previously[9–12] simply because it is longer than usual in order to account for the protracted mossy fiber discharges observed in vivo[32,33]. All recorded GrCs were silent at rest and their response frequency at 500 ms showed little or no adaptation (Fig. 1a). However, surprisingly enough, while initial GrC responses corresponded to the classical description, at longer times they showed a richness of different properties (Fig. 1a). In some cells firing remained stable (non-adapting, 20.6%), in others it slowed-down or even stopped (adapting, 66.7%), while yet in others it increased (accelerating, 12.7%). While changes in the initial 500 ms were less than about ±20%, changes over the whole 2000 ms time-window could exceed ±80% (Fig. 1b).

The cerebellar GrCs normally show a linear firing frequency increase with current injection[9–12]. During the first 500 ms of the response, both in non-adapting, adapting and accelerating GrCs, the firing frequency increased monotonically and almost linearly with current injection ($f_{initial}/I$ plots) (Fig. 1c).

In summary, when observed during the first 500 ms of the response to current injection, GrC firing frequency was almost stable and the input–output relationships linear, conforming to general knowledge. However, a richness of electroresponsive properties emerged at longer response times (here up to 2000 ms).

The firing frequency changes occurring over 2000 ms current steps were assessed by calculating the intrinsic frequency change $IFC = [(f_{final} − f_{initial})/f_{initial}]\%$, in which $f_{initial}$ and $f_{final}$ are the spike frequencies at the beginning and end of current injection. In this equation, adaptation and acceleration are characterized by $IFC < 0$ and $IFC > 0$, respectively, while $IFC = 0$ occurs in the absence of changes. Characteristically, the accelerating GrC showed an IFC-positive peak around 10 pA current injection, while the other GrCs showed negative IFC values (see Fig. 1d).

### Different synaptic excitation properties at the MF-GrC relay. The different firing properties of GrCs could have an impact on transmission of MF discharges. This issue was addressed by stimulating the MF bundle at frequencies $f_{stim} = 5–100$ Hz and measuring the GrC response frequency, $f_{resp}$. GrCs showed different transmission properties, from one-to-one responses over the whole-input frequency range to responses faster or slower than the input (Fig. 2a). These properties were analyzed using SFC/$f_{stim}$ plots, where $SFC = [(f_{resp} − f_{stim})/f_{stim}]\%$ is the synaptic frequency change (Fig. 2b). The SFC/$f_{stim}$ plots showed distinctive trajectories (Fig. 2b). The accelerating GrCs showed $SFC > 0$ (enhanced output) around 20 Hz, while the other GrCs showed $SFC < 0$ all over the frequency range.

Since MF-GrC transmission involves high-frequency bursts[29], we evaluated the MF-GrC signal/noise ratio, $S/N$ (Fig. 2c). This was measured when a signal (high-frequency burst at 100 Hz) was delivered over noise (background activity at 20 Hz) (see Fig. 2c). The adapting GrCs suppressed the background efficiently but allowed high-frequency burst transmission resulting in high $S/N$. The non-adapting and accelerating GrCs showed less-efficient background suppression resulting in lower $S/N$. Thus, GrCs with different adaptation/acceleration properties also showed differential filtering of MF activity.

In adapting and accelerating GrCs, a voltage-clamp protocol was run to monitor the effectiveness of synaptic stimulation. The average EPSC amplitude was –39.5 ± 4.8 pA ($n = 24$), corresponding to activation of ~2 synapses on average (e.g., cf. refs. [34,39]). The paired-pulse ratio (PPR) in 5–100 Hz trains was 0.88 ± 0.05 in strong-adapting GrCs ($n = 4$ cells, 47 measures) and 0.53 ± 0.1 in accelerating GrCs ($n = 5$ cells, 46 measures). By

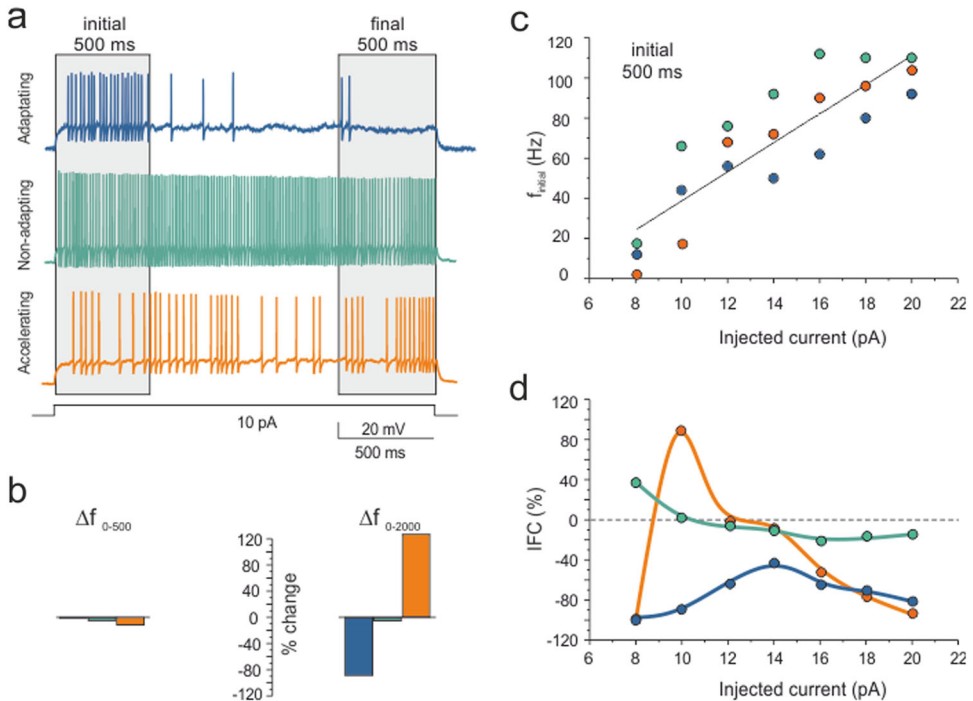

**Fig. 1 Different GrC properties: adaptation and acceleration.** Three exemplar GrC recordings are shown, one adapting, one non-adapting, one accelerating. The same color codes are used consistently in the figure. **a** Voltage responses to 2000 ms–10 pA current injection from the holding potential of –65 mV. Spike frequency initially remains stable in all the three cells but it shows different trends thereafter. **b** $\Delta f_{0-500}$ and $\Delta f_{0-2000}$ are the spike frequency % changes after 500 ms and 2000 ms, respectively. **c** In $f_{initial}/I$ plots, spike frequency increases almost linearly with the injected current intensity in both the accelerating, adapting, and non-adapting GrCs ($R^2 = 0.90$, $p = 0.0007$). **d** Plot of the intrinsic frequency change IFC vs. injected current for the three GrCs (each line represents a GrC with the same color code as in **a**). A positive peak is apparent in the accelerating GrC at 10 pA current injection, while negative IFC values prevail in the other GrCs.

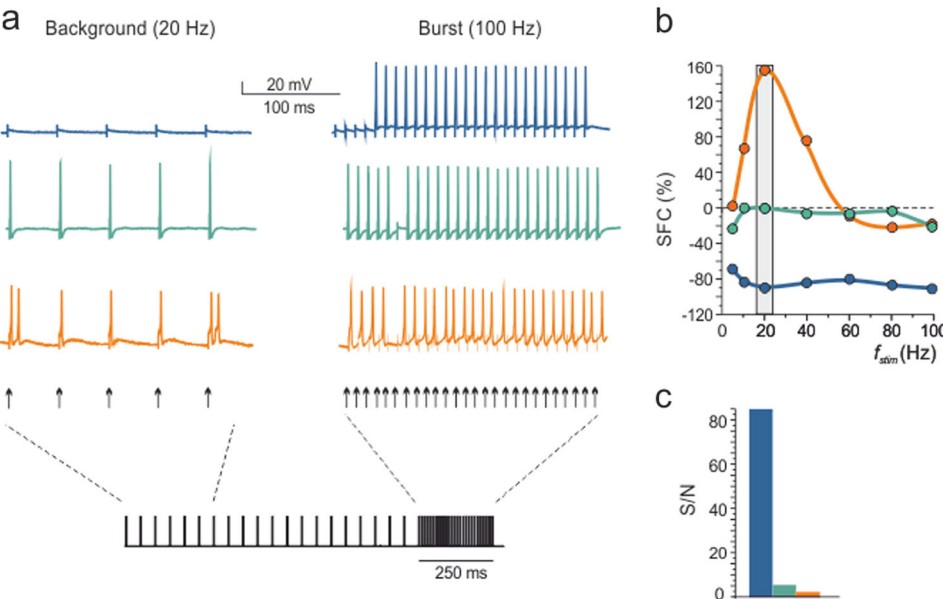

**Fig. 2 Different MF-GrC properties: synaptic responsiveness and S/N.** Three exemplar GrC cell recordings are shown, one adapting, one non-adapting, one accelerating, as defined by their intrinsic electroresponsiveness (same cells and color codes as in Fig. 1). **a** Voltage responses of exemplar GrCs to electrical stimulation of the MF bundle (1 s continuous stimulation at 20 Hz followed by 250 ms at 100 Hz) from the holding potential of –65 mV. **b** Plot of the synaptic frequency change SFC vs. stimulus frequency for the three GrCs shown in **a** (each line represents a GrC with the same color code as in **a**). A positive peak is apparent in the accelerating GrC at 20 Hz background stimulation, while negative SFC values prevail in the other GrCs. **c** Signal-to-noise ratio, ($S/N$ = fresp@100 Hz/fresp@20 Hz) for the three GrCs in **a**, **b**. Note that $S/N$ is much higher in adapting that in the other two GrCs.

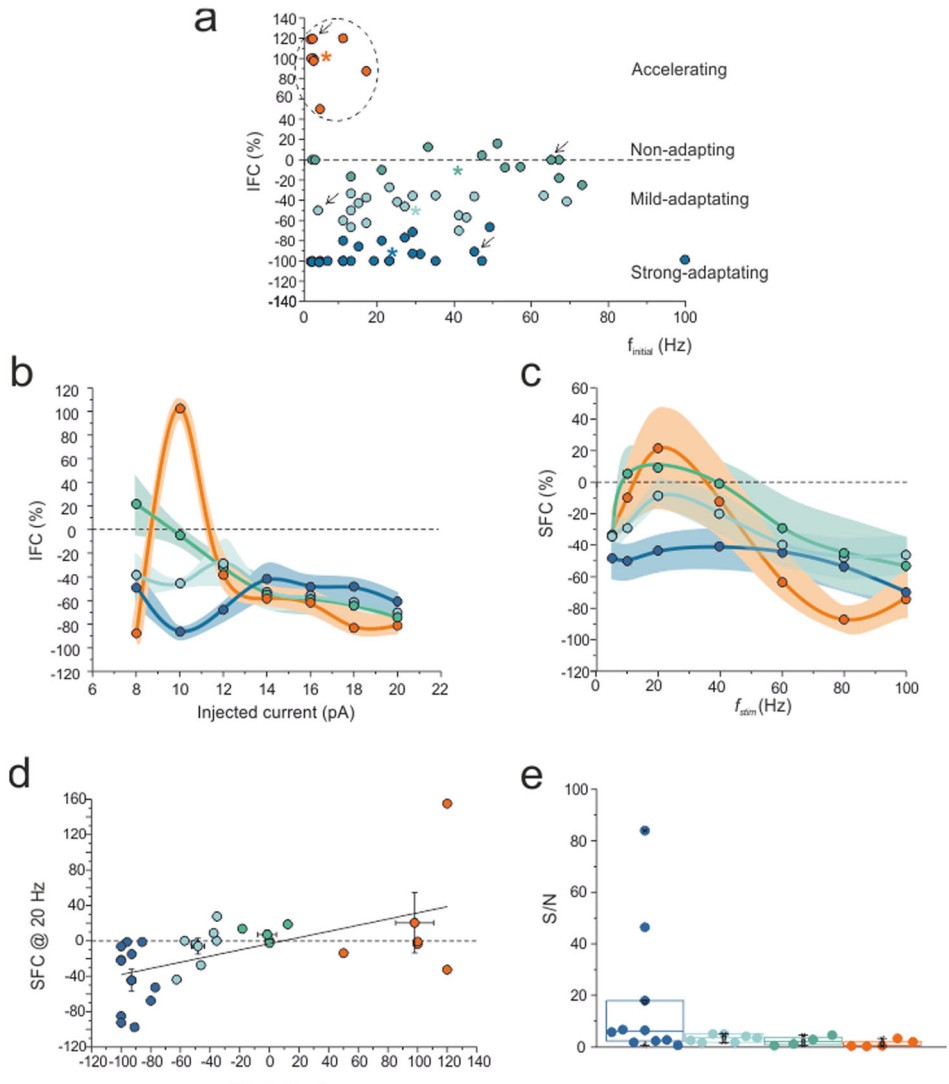

**Fig. 3 Average properties: GrC subtypes. a** $k$-means cluster-analysis applied to the non-accelerating GrCs ($n = 55$ at 10 pA) using IFC and finitial as features. $k$-means identifies three statistically different data clusters ($p < 0.05$; one-way ANOVA test). The cluster of accelerating GrCs was well separated and identified by its peculiar electrophysiological activity (dotted circle). The centroids of the four subpopulations are indicated by asterisk (*). **b, c** The cells identified in the four clusters were used to construct average IFC/I and SFC/$f_{stim}$ plots (average ± SEM). Note, in both graphs, the similar and progressive change of properties from strong-adapting to mild-adapting, non-adapting, and accelerating GrCs. **d** IFC@10 pA and SFC@20 Hz reveal a linear correlation ($R^2 = 0.70$). The average ± SEM is shown for each cell custer. **e** The cells identified in the four cluster are used to construct $S/N$ box-and-whisker plots ($S/N = fresp@100$ Hz/fresp@20 Hz). Note the progressive decrease of $S/N$ from adapting to accelerating GrCs. Color codes as in Fig. 1.

comparison with the precise determinations carried out on this same synapses[34], these estimates suggested that adapting GrCs were activated by synapses with lower release probability than accelerating GrCs ($n = 93$, $p = 0.0036$, unpaired $t$-test).

**Average properties of GrC subtypes.** In order to determine whether the different GrC responses represented a continuum or rather could be grouped into subpopulations, an unbiased $k$-means cluster-analysis was applied to the population of non-accelerating GrCs ($n = 55$ at 10 pA) using IFC and $f_{initial}$ as features (Fig. 3a). The $k$-means analysis was performed at low-current intensity, where differences among GrCs were more evident. The $k$-means analysis identified three statistically different data clusters corresponding to strong-adapting GrCs ($n = 23$), mild-adapting GrCs ($n = 19$), and non-adapting GrCs ($n = 13$) (the adapting GrCs were actually subdivided into two groups). The accelerating GrCs ($n = 8$) were already well indentified by their peculiar electrophysiological

behavior, and actually their inclusion into $k$-means resulted in a new cluster without altering the distribution of GrCs among strong-adapting, mild-adapting, and non-adapting subtypes. (Fig. 3a) (see "Methods" for details). The cells identified in the four clusters were then used to analyze their average properties.

In the average IFC/I plot (Fig. 3b), non-adapting, mild-adapting and strong-adapting GrCs showed differential adaptation at low-current injection (<14 pA) but converged toward a similar adaptation level at high-current injection (20 pA). The accelerating GrCs showed increased intrinsic electroresponsiveness at low-current injection (around 10 pA), but decreased it to the level of the other GrCs at higher current injections (20 pA).

In the average SFC/$f_{stim}$ plot (Fig. 3c), non-adapting, mild-adapting, and strong-adapting GrCs showed differential response regimens at low-synaptic stimulation frequencies (<50 Hz), but converged toward a similar response level at higher synaptic stimulation frequencies (100 Hz). The accelerating GrCs showed

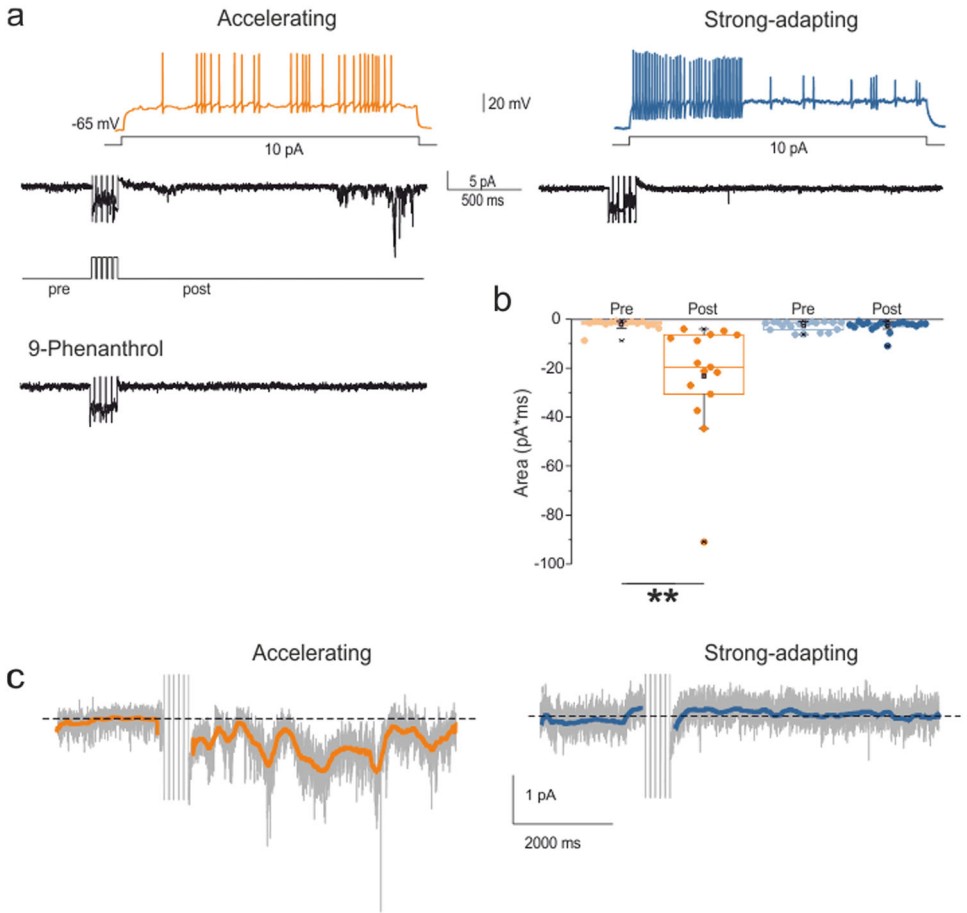

**Fig. 4 TRPM4 channels in accelerating GrCs. a** The voltage traces show the GrC response to a 10 pA step current in an accelerating and strong-adapting GrCs. A five-steps depolarizing protocol is applied to uncover DISC. In the accelerating GrC, DISC typically appears as a delayed burst of rapid events. DISC cannot be longer elicited after perfusing a specific TRPM4 channel blocker, 9-Phenanthrol (100 μM). No DISC is visible in the strong-adapting GrC (all currents low pass filtered at 500 Hz). **b** The average area of DISC transients shows a significant increase after the depolarizing protocol in accelerating GrCs (**$p = 0.02$, $n = 15$, paired $t$-test,) but not in strong-adapting granule cells. **c** Raw average current traces for accelerating and strong-adapting GrCs are shown in light gray. In accelerating GrCs ($n = 15$) the DISC currents are still evident at 4 s, while no net current changes are visible in strong-adapting GrCs even 6 s after the impulse train ($n = 27$). The corresponding filtered average traces (orange and blue) are shown superimposed (adjacent-averaging smoothing).

increased synaptic responsiveness at low-input frequencies (around 20 Hz), but decreased it to the level of the other GrCs at higher input frequencies (100 Hz).

The average IFC and SFC plots showed similar trends for the four GrC categories. The correlation between IFC and SFC (Fig. 3d) was evaluated in a characteristic point corresponding to the peak of accelerating GrCs, i.e., using IFC at 10 pA and SFC at 20 Hz. The *IFC@10 pA/SFC@20 Hz* plot actually revealed a linear correlation ($R^2 = 0.70$).

*S/N* was also evaluated at the cell population level in the four GrC clusters identified by *k*-means analysis (Fig. 3e). This analysis revealed that *S/N* was indeed progressively lower when passing from strong-adapting to mild-adapting, non-adapting, and accelerating GrCs.

**TRP current expression in accelerating GrCs**. The depolarization-induced slow current (DISC) is a depolarizing current gated by the raise of intracellular $Ca^{2+}$ that follows action potential bursts and NMDA channel activation, and can typically generate a secondary burst after 1.5–2 s like that observed in accelerating GrCs[19,20]. DISC is typically generated by TRPM4[21], a $Ca^{2+}$-activated non-selective cation channel, which provides a strong depolarizing drive upon

$Ca^{2+}$ entry[22]. The presence of TRPM4 currents was explored in a separate set of recordings using the same voltage-clamp protocol adopted in Purkinje cells (PCs)[21] (Fig. 4a), which indeed elicited TRPM4-like currents in accelerating GrCs ($n = 15$ out of 15). These currents occurred after $1341.6 \pm 245.9$ ms and were typically burst-like with a maximum charge transfer of $-23.3 \pm 5.8$ pA*ms ($n = 15$) within 2 s from the impulse train (Fig. 4b). In order to confirm TRPM4 specificity, the preparations were perfused with 100 μM 9-Phenanthrol, a specific TRPM4 channel blocker[21,23], which blocked the currents ($n = 9$ out of 9). The spike discharge recorded after switching to current-clamp in these same accelerating GrCs showed increased frequency in correspondence to the TRPM4 current, with *IFC@10 pA* $= 96.3 \pm 10.2\%$ (Fig. 4a). Conversely, non-accelerating GrCs never showed any TRPM4-like current ($n = 27$ out of 27). Indeed, no net current changes were visible in strong-adapting GrCs even after 6 s from the impulse train, whereas the average TRPM4 current in the ten accelerating GrCs was still about –1 pA after 4 s (Fig. 4c).

A recent investigation demonstrated that TRPM4 protein is abundantly expressed in cerebellar PCs, while its distribution in the granular layer was not evaluated[21]. However, according to the in situ hybridization data from the Allen Institute of Brain Science (http://mouse.brain-map.org/), TRPM4 channels should also be

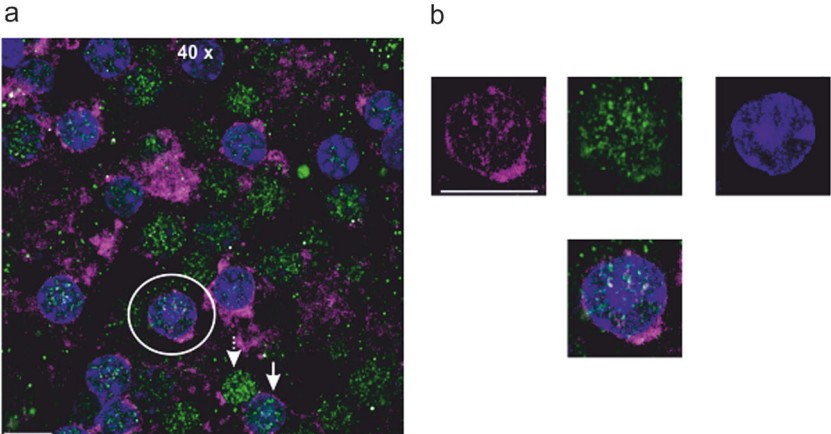

**Fig. 5 TRPM4 expression in GrCs. a** Fluorescence image of a slice cut from the cerebellar vermis (220 μm thick) treated with polyclonal antibodies directed against TRPM4 (magenta) and PAX6 (green) and stained with Hoechst to visualize cell nuclei (blue). TRPM4 expression is visible as a thin-outline around GrCs membrane, while PAX6 and Hoechst fluorescence are confined to the nuclear region (dotted arrow). Note the typical rounded shape of the GrCs and the nucleus occuping most of the cytoplasm. Alternatively, either PAX or Hoechst signals dominate nuclear fluorescence (arrows show two examplar cells) (x40 objective magnification, scale bar 5 μm). **b** A GrC was selected from **a** (white circle) and shown at x2 magnification (scale bar 10 μm). The upper panels show the acquisition channels separately (TRPM4, PAX6, and Hoechst, respectively), while in the bottom panel the three channels are merged to highlight the coexpression of TRPM4, PAX6, and Hoechst. It should be noted that TRPM4 staining surrounds the nucleus and that some dots can also be observed inside the cell.

expressed in GrCs. To validate this preliminary information at protein level, we carried out immunohistochemistry on the rat cerebellar granular layer by using a rabbit polyclonal antibody raised against the amino acid residues 5–17 of human TRPM4 (Fig. 5a). GrCs were identified by co-staining the slices with Hoechst and PAX6, a GrC-specific protein[40]. Confocal microscopy revealed that TRPM4 protein was expressed in most GrCs both on the plasma membrane and within the cytosol (Fig. 5b). This subcellular pattern of expression is similar to that reported in other brain areas[24,25].

**Modeling predicts parameter tuning in GrC subtypes.** Since it is experimentally unpractical to determine the balance of multiple ionic conductances in single GrCs, we inferred membrane mechanisms from simulations using biophysically detailed data-driven models incorporating multiple types of ionic channels on the dendrites, soma, hillock, axonal initial segment (AIS), ascending axon (AA), and parallel fibers (PFs) (Fig. 6a; see also Supplementary Fig. 1)[13–17]. The hypothesis that the known set of ionic channels was indeed sufficient to explain the GrC firing subtypes was explored using automatic optimization of maximum ionic conductances ($G_{max}$)[17] yielding a family of solutions that fit the experimental "template" (Fig. 6b).

The models were first optimized, without coupling TRPM4 channels, toward a template taken from the first 500 ms discharge in a non-adapting GrC. All the models could faithfully reproduce the stable regular firing behavior typical of the first 500 ms of discharge (Fig. 6b) while, interestingly, adapting properties emerged at later times. The coupling of TRPM4 channels to $Ca^{2+}$ through Calmodulin allowed to obtain accelerating GrCs (Fig. 6b). Therefore, the ability to generate adaptation and acceleration was intrinsic to the ionic channel complement through fine-tuning of $G_{max}$ values (see Supplementary Fig. 2) and TRPM4 coupling.

In the models, the MF-GrC synapse was implemented using a dynamic representation of the vesicle cycle, that could faithfully reproduce MF-GrC short-term plasticity, including EPSC depression and facilitation[16,41–44]. The different GrC models were stimulated synaptically with a protocol identical to that used for experimental recordings. According to experimental estimates

(see above and refs. [34,45]), the GrC model was activated by two synapses with lower release probability in the strong-adapting GrC ($p = 0.1$) than in the other GrCs subtypes ($p = 0.5$). The GrC model responses at different frequencies, as well as the SFC/$f_{stim}$ plots and the $S/N$ values were remarkably similar to those obtained experimentally (Fig. 6c).

As a whole, GrC and MF models parameter tuning yielded electroresponsive and synaptic transmission properties that closely matched those observed experimentally (see Supplementary Fig. 3). Moreover, since the models effectively captured the phenomenological properties of GrC subtypes, they were further used to infer the underlying ionic mechanisms.

**Mechanisms generating firing adaptation and acceleration.** Adaptation: in the model, the larger Cav2.2 maximum conductance in strong-adapting than non-adapting GrCs (Fig. 7a) caused larger $Ca^{2+}$ currents, which enlarged the spike upstroke by 5–10 mV, compatible with the effects of specific N-type $Ca^{2+}$-channel blockers observed in cerebellar slices[12] (Fig. 7b). The larger upstroke enhanced the activation of $Ca^{2+}$-dependent and voltage-dependent $K^+$ channels, increasing $K^+$ currents and spike AHP. The larger AHP, in turn, enhanced de-inactivation of the A-type current, which is known to protract the ISI[46]. Likewise, a large AHP favored M-type current deactivation/reactivation, which can effectively slow-down (or even block) firing for hundreds of ms[47]. As a whole, the model predicted that a primary increased of $Ca^{2+}$ currents would cause a subsequent increase of $K^+$ current by ~8 pA in the ISI capable of explaining firing adaptation (Fig. 7b; see also Supplementary Figs. 4a, 5, 6).

Acceleration: the TRPM4 channel, which was activated by spike trains, caused a sizeable $Ca^{2+}$ influx through $Ca^{2+}$ channels (Fig. 7c). Cooperative $Ca^{2+}$ binding to Calmodulin generated a CaM2C complex that gated TRPM4 channels in a non-linear manner causing their opening once a critical concentration threshold was reached. The consequent inward current depolarized the membrane, thereby accelerating firing (see also Supplementary Fig. 4b).

**Mechanisms differentiating synaptic responsiveness.** EPSC trains at 100 Hz were simulated using different release probabilities

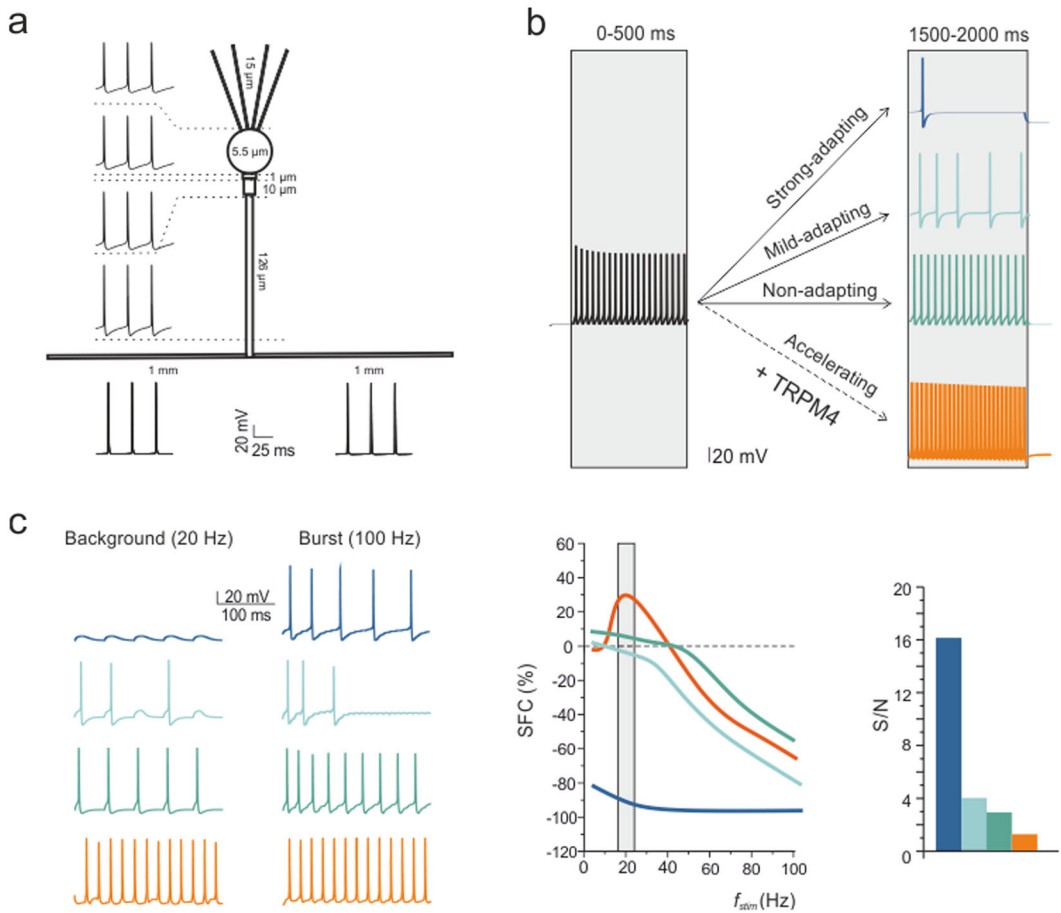

**Fig. 6 Modeling GrC and MF-GrC richness of properties. a** Schematic representation of GrC morphology (not to scale). The spike shape matches experiments. The ionic channels type and distribution are reported in Supplemental Material. **b** An example of model trace obtained using a current injection of 10 pA. The optimization is run over the first 500 ms of discharge of a GrC yielding a family of solutions with maximum conductances falling within the physiological range of maximum conductance values. At later times (1500–2000 ms), the discharge patterns diverge into non-adapting, mild-adapting, and strong-adapting solutions. Coupling of the $Ca^{2+}$/Calmoldulin/TRPM4 mechanism to the model yields accelerating GrCs. **c** Voltage responses of GrC models to activation of the MF-GrC synapse (two synapses, 1 s continuous stimulation at 20 followed by 250 ms at 100 Hz) from the holding potential of –65 mV. The $p$-values are: strong-adapting GrC ($p = 0.1$), mild-adapting, non-adapting, and accelerating GrC ($p = 0.5$). The synaptic frequency change SFC vs. stimulus frequency show a positive peak in the accelerating GrC model at 20 Hz background stimulation, while negative SFC values prevailed in the other GrC models. The Signal-to-noise ratio, ($S/N = $ fresp@100 Hz/fresp@20 Hz) is much higher in strong-adapting GrC than in the others. Color codes as in Fig. 3.

($p = 0.1, 0.5, 0.9$) (Fig. 8a) and used to calculate the corresponding PPR[34]. The PPR/$p$ plot showed a negative slope, such that higher PPR corresponded to lower $p$-values (Fig. 8b; see also Supplementary Fig. 7). A projection of experimental PPR values to corresponding $p$-values through the PPR/$p$ plot yielded $p = 0.43 \pm 0.06$ for strong-adapting GrCs and to $p = 0.85 \pm 0.12$ for accelerating GrCs. It should be noted that, by considering the whole PPR data distributions, $p$-values in strong-adapting GrCs could range down to 0.1 and those in accelerating GrCs range up to 1.

Since $p$ is a main factor regulating synaptic integration and excitation in response to input trains, the effect of different $p$-values ($p = 0.1, 0.5, 0.9$) was tested together with different numbers of active synapses (1–4) to systematically explore the SFC and $S/N$ space. The mean SFC and $S/N$ values for the four GrC groups are reported in the three-dimensional graph of Fig. 8c, d. Whatever the number of active synapses, SFC at 20 Hz was larger for accelerating than strong-adapting GrCs. The low SFC and high $S/N$ values (>10) typical of strong-adapting GrCs were found at low release probability ($p = 0.1$), while the high SFC and low $S/N$ values (<5) typical of all the other GrCs types were found at high release probabilities ($p = 0.5–0.9$) in accordance to initial estimates

derived from PPR analysis. Therefore, the model predicts that high $S/N$ ratios typical of strong-adapting GrCs can be expected at low $p$, matching experimental $p$ determinations.

## Discussion

This paper shows that cerebellar GrCs, in contrast to the canonical view describing them as a homogeneous population of neurons generating regular firing, actually show a rich repertoire of firing patterns. Over prolonged discharges (~2 s), some GrCs remain non-adapting (20.6%) but others show adaptation to various degree (66.7%) or, conversely, acceleration (12.7%). Adaptation and acceleration are predicted to reflect fine-tuning of membrane ionic conductances and the activation of a previously undisclosed TRPM4 channel. Specific neurotransmission properties further differentiate synaptic responsiveness.

A $k$-means classifier based on cerebellar GrC discharge properties uncovered four different subtypes based on intrinsic electroresponsiveness: strong-adapting, mild-adapting, non-adapting, and accelerating GrCs. Interestingly, these properties reverberated into the response to MF stimulation (IFC/SFC plots showed a

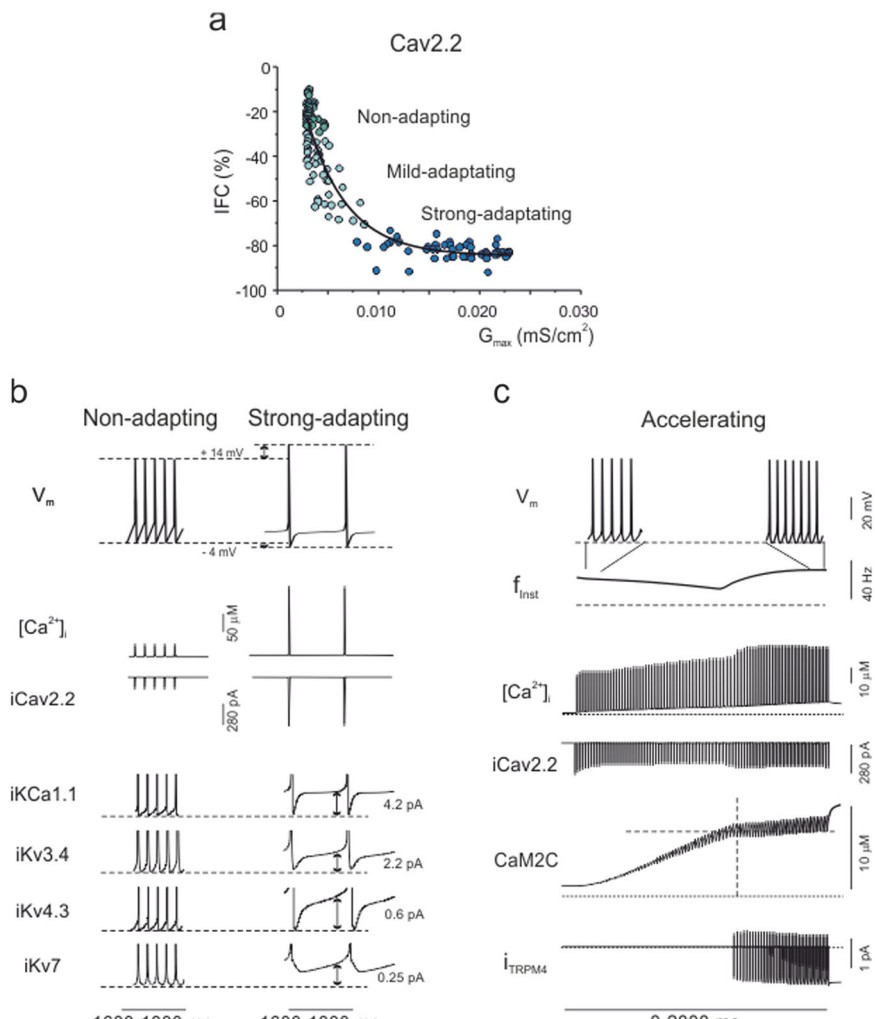

**Fig. 7 Prediction of the ionic mechanisms of firing adaptation and acceleration. a** The graph shows the distribution of IFC in GrC models with respect to the Cav2.2 maximum conductance (mS/cm$^2$), revealing a negative correlation ($R^2 = 0.88$, $n = 150$, $p < 10^{-4}$). At the level of GrC subtypes, there is a clear unbalance of Cav2.2 maximum conductance in favor of adapting GrCs. **b** The traces show the voltage and intracellular Ca$^{2+}$ concentration along with ionic currents in non-adapting and strong-adapting GrCs during 10 pA current injection at 1700 ms. Double arrows measure the difference between non-adapting and strong-adapting GrCs. A scheme showing the hypothetical mechanisms differentiating the two cell subtypes is reported in Supplemental Material. **c** The traces show the voltage and intracellular Ca$^{2+}$ concentration along with ionic currents and elements of the intracellular coupling mechanism in an accelerating GrCs during 10 pA current injection. A scheme showing the chemical reactions, leading to TRPM4 channel opening is reported in Supplemental Material.

linear correlation, see Fig. 3d). Since MFs convey combinations of frequency-modulated spike trains and bursts[29–33], we evaluated how well GrCs could discriminate bursts from long-train discharges by estimating *S/N*. *S/N* was much larger in strong-adapting than in other GrC subtypes and this turned out to depend not only on different intrinsic electroresponsiveness but also on different release probability, *p*. According to PPR analysis[34], *p* was lower in strong-adapting than in the other GrC subtypes. Modeling showed that, at low release probability (e.g., $p = 0.1–0.4$), ensuing short-term facilitation could prevent low-frequency background transmission, while still allowing transmission of high-frequency bursts. Conversely, at high release probability (e.g., $p = 0.5–0.9$), ensuing short-term depression allowed similar transmission of both background and bursts (cf. ref. [16]). Therefore, intrinsic discharge properties and synaptic tuning concurred in differentiating the properties of synaptic excitation supporting the existence of functional MF-GrC channels specialized for differentiated signal processing.

Adaptation reflected high values of the high-threshold Ca$^{2+}$ conductance, which increased the Ca$^{2+}$ current and raised the spike overshoot[12]. This brought about a stronger activation of voltage and Ca$^{2+}$-dependent K$^+$ currents deepening the undershoot and increasing A-type and M-type K$^+$ currents. As a whole, the K$^+$ current increased by ~8 pA during the ISI of strong-adapting GrCs explaining firing slow-down. Acceleration was correlated with the TRPM4 current (~1 pA) and modeling predicted that the TRPM4 channels, coupled to Calmodulin through intracellular Ca$^{2+}$ changes, could effectively generate firing acceleration about 1.5–2 s after the beginning of discharge, i.e., when acceleration was observed experimentally[19–21]. This delay reflected the slow cooperative gating of TRPM4 channels, that opened only after the Ca$^{2+}$-Calmodulin complex reached a threshold. Simulations therefore predict that fine-tuning of Ca$^{2+}$ influx and coupling to TRPM4 channels would be critical for determining the difference between GrCs showing adaptation or acceleration[48]. It should be noted that, both in the case of

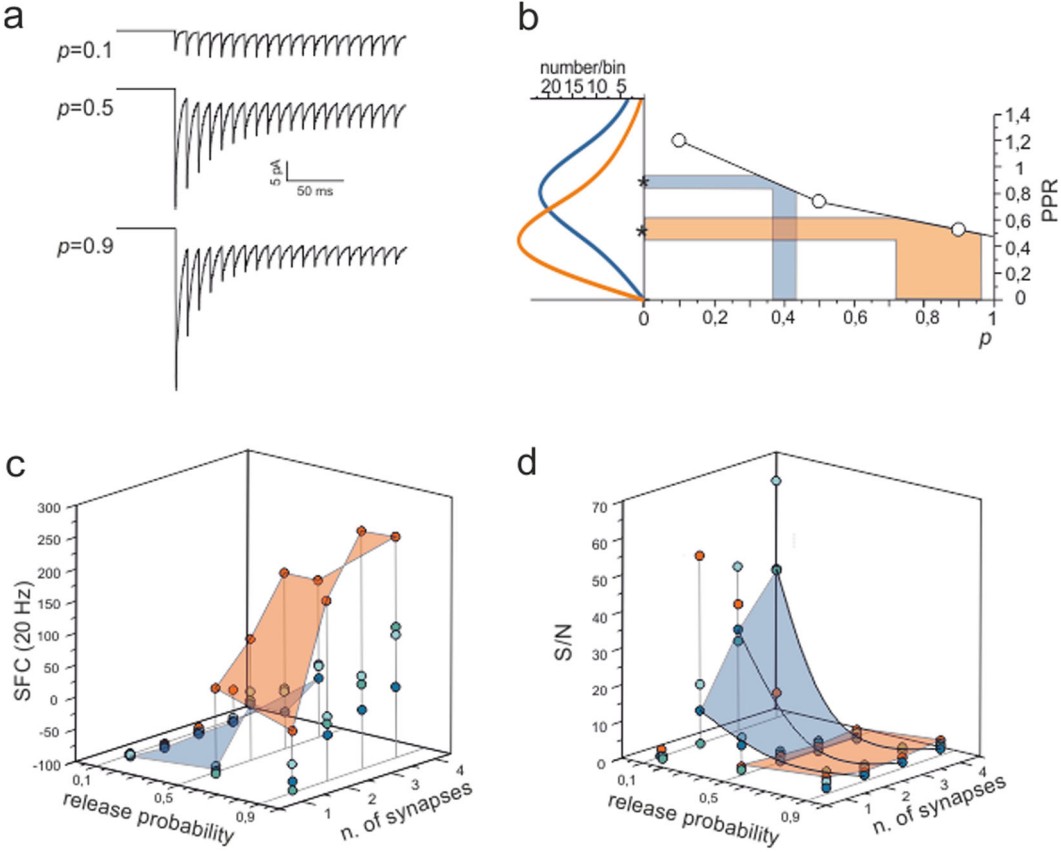

**Fig. 8 Prediction of the mechanisms of synaptic responsiveness and S/N. a** The traces show voltage-clamp simulations of synaptic activity using MF-GrC synapses with a release probability $p = 0.1$, 0.5, and 0.9. Two synapses area activated in each simulations. **b** Prediction of $p$-values from PPR data using model simulations. Asterisk (*) indicate mean experimental values of release probability in accelerating and strong-adapting GrCs, and the colored bands correspond to SEM. The probability density functions of experimental PPRs are reported on the left axis. The predicted PPR values at different $p$-values are shown as open circles connected by straight lines. Through this model curve, the experimental PPRs are reflected into predicted $p$-values. **c** Model prediction of SFC as a function of $p$ and of the number of active synapses for a 20 Hz stimulation. Note that SFC is higher for accelerating than for all other GrC models at high $p$. **d** Model prediction of $S/N$ ($S/N = $ fresp@100 Hz/fresp@20 Hz) as a function of $p$ and of the number of active synapses. Note that $S/N$ is higher in strongly adapting GrC models at low $p$.

acceleration and adaptation, the very high input resistance of GrCs (GΩ range) amplified the effect of pA-range modulating currents like those considered here[12].

TRPM4 is a $Ca^{2+}$-dependent non-selective cation channel that is equally permeable to $Na^+$ and $K^+$ but impermeable to $Ca^{2+}$ (ref. [22]). In excitable cells, it is regarded as the most suitable signaling mechanism to boost depolarization following an increase in electrical activity, which drives the activation of voltage-gated $Ca^{2+}$ channels[22]. Emerging evidence shows that TRPM4 is actually recruited by an increase in cytosolic $Ca^{2+}$ concentration to generate a DISC current and fine-tune neuronal excitability in several brain areas[19,24,26]. For instance, TRPM4 mediates the depolarizing afterpotential and phasic bursting observed in supraoptic and periventricular nuclei of the hypothalamus after a train of action potentials[25] and contributes to increase the firing rate in PCs[21]. Here, DISC currents were first recorded in cerebellar GrCs and their TRPM4-mediated nature was revealed by the blocking effect of 100 μM 9-Phenanthrol. Likewise, 100 μM 9-Phenanthrol allowed to identify TRPM4 currents in several other neurons, including cerebellar Purkinje cells[21] and mouse prefrontal cortex neurons[24,26].

There are several possible reasons for why DISC current and firing acceleration were observed in just ~12% of GrCs. Apparently, immunoistochemistry showed different patterns of TRPM4 and PAX6 (related to GrC development[40]) expression and of Hoechst staining (related to chromatin folding) in GrCs (cf. Fig. 5a), which may reflect different functional states of the neurons. One can further speculate that the actual activation of TRPM4 depends on its membrane expression and physical coupling with transduction cascades involving Cav2.2, e.g., on whether a threshold sub-membrane $Ca^{2+}$ concentration is reached upon the initial discharge next to TRPM4 channels. Another cue is provided by the peculiar kinetics of the TRPM4 current, which, unlike in Purkinje cells[21], consisted of transient bursts (cf. Fig. 4a). These bursts strongly resemble those induced by intracellular $Ca^{2+}$ release through ryanodine receptors (RyRs) and inositol-1,4,5-trisphosphate (InsP₃) receptors in pancreatic β-cells[49]. This implies that TRPM4-mediated currents in GrCs would involve delayed endoplasmic reticulum $Ca^{2+}$ spikes induced by extracellular $Ca^{2+}$ entry through Cav2.2 opening. Although TRPM4 was sufficient to explain firing acceleration, we cannot exclude that other TRP channels, such as TRPM5 and TRP Canonical 5 (TRPC; e.g., see ref. [18]) could also contribute. Future experiments may exploit knockout mice to dissect the contribution, if any, of other $Ca^{2+}$-dependent conductances.

The richness of GrC intrinsic and synaptic responsiveness ends up in two main transmission patterns. In accelerating, non-adapting and mild-adapting GrCs, all frequencies are transmitted faithfully, with the accelerating GrCs being especially suitable to maintain reliable transmission at low frequencies. In strong-adapting GrCs,

the low frequencies are suppressed while the high-frequencies are transmitted, so that these neurons operate as high-pass filters. These transmission properties suggest that GrCs can process incoming MF inputs through multiple frequency-dependent filters, akin with the theoretical prediction of the adaptive filter model (AFM)[38,50]. It is tempting to speculate that there are MF-GrC channels specialized for different input patterns. This specialization may be the result of plasticity causing pre- and post-synaptic changes rewiring the system and optimizing signal transfer (e.g., LTD may characterize low-$p$ MFs synapses with strong-adapting GrCs, while LTP may characterize high-$p$ MFs synapses with accelerating GrCs)[34,45]. Any potential relationships between these putative transmission channels and zebrin stripes[51] and GrC functional clusters[52] remains to be determined.

Fine-tuning of postsynaptic ionic conductances allowed the diversification of cerebellar GrC firing patterns. Beyond that, the balance of these conductances matched specific setting of presynaptic neurotransmitter release probability. Thus, what may simply be regarded as biological variability or noise turns out, in fact, into a richness of properties that the circuit could exploit to carry out its internal computations[1–3]. Similar considerations may apply to other neurons like those of the hippocampus[53]. A question that remains to be answered is now whether differentiated neuronal and synaptic properties are determined by neuromodulatory processes or induced by plasticity. For instance, plasticity of synaptic transmission and intrinsic excitability, which have been observed in GrCs[16,36,54], may exploit ionic channel and synaptic parameter tuning causing the differentiation of GrC subtypes. Moreover, it would be important to determine how these properties are spatially distributed inside the cerebellar circuit thereby generating specific transmission channels shaping spatio-temporal recoding and adaptive filtering of incoming spike trains[5,6,50].

## Methods

**Experimental methods**. All experimental protocols were conducted in accordance with international guidelines from the European Union Directive 2010/63/EU on the ethical use of animals and were approved by the ethical committee of Italian Ministry of Health (639.2017-PR; 7/2017-PR).

Slice preparation and solutions: cerebellar GrCs were recorded from the vermis central lobe of acute parasagittal cerebellar slices (230 μm thick) obtained from 18- to 24-day-old Wistar rats of either sex. Slice preparation and patch-clamp recordings were performed as reported previously[9,12,55]. Briefly, rats were decapitated after deep anesthesia with halothane (Sigma, St. Louis, MO), the cerebellum was gently removed and the vermis was isolated, fixed on a vibroslicer's stage (Leica VT1200S) with cyano-acrylic glue and immersed in cold (2–3 °C) oxygenated Kreb's solution containing (mM): 120 NaCl, 2 KCl, 2 CaCl₂, 1.2 MgSO₄, 1.18 KH₂PO₄, 26 NaHCO₃, and 11 glucose, equilibrated with 95% O₂–5% CO₂ (pH 7.4). Slices were allowed to recover at room temperature for at least 40 min before being transferred to a recording chamber mounted on the stage of an upright microscope (Zeiss, Germany). The slices were perfused with oxygenated Krebs solution (2 mL/min) and maintained at 32 °C with a Peltier feedback device (TC-324B, Warner Instrument Corp., Hamden, CT, USA).

Patch-clamp recordings and analysis: whole-cell patch-clamp recordings from cerebellar GrCs ($n = 63$) were performed with Multiclamp 700B [-3dB; cutoff frequency (fc), 10 kHz], sampled with Digidata 1440A/1550 interface, and analyzed off-line with pClamp10 software (Molecular Devices, CA, USA), MS Excel, Matlab (Mathworks, Natick, MA) and OriginPro software.

Patch-clamp pipettes were pulled from borosilicate glass capillaries (Hilgenberg, Malsfeld, Germany) and had a resistance of 7–9 MΩ before seal formation when filled with the intracellular solution containing (in mM): 126 potassium gluconate, 4 NaCl, 5 Hepes, 15 glucose, 1 MgSO₄.7H₂O, 0.1 BAPTA-free, 0.05 BAPTA-Ca²⁺, 3 Mg²⁺-ATP, 0.1 Na⁺-GTP, pH 7.2 adjusted with KOH. The Ca²⁺ buffer was estimated to maintain free Ca²⁺ concentration around 100 nM. Just after obtaining the cell-attached configuration, electrode capacitance was carefully canceled to allow for electronic compensation of pipette charging during subsequent current-clamp recordings. The stability of whole-cell recordings can be influenced by modification of series resistance ($R_s$). To ensure that $R_s$ remained stable during recordings, passive cellular parameters were extracted in voltage-clamp mode by analyzing current relaxation induced by a 10 mV step from a holding potential of –70 mV. The transients were reliably fitted with a bi-exponential function yielding membrane capacitance ($C_m$) of 3.1 ± 0.1 pF, membrane resistance ($R_m$) of 1.1 ± 0.1 GΩ, and

series resistance ($R_s$) of 22.8 ± 1.1 MΩ. The –3 dB cell plus electrode cutoff frequency, $f_{VC} = (2\,R_s C_m)^{-1}$, was 2.6 ± 0.1 kHz ($n = 63$)[9,56,57].

GrCs intrinsic excitability was investigated in current-clamp mode by setting resting membrane potential at –65 mV and injecting 2 s current steps (from –8 to 22 pA in 2 pA increment). The action potentials frequency was computed in two time windows of the duration of 500 ms, (0–500 ms and 1500–2000 ms, respectively). In a subset of experiments ($n = 26$) the MF bundle was electrically stimulated to investigate differences in synaptic transmission. The stimulation was performed with a large-tip (10–20 μm) patch-pipette filled with extracellular Krebs solution, via a stimulus isolation unit. The stimulation protocol comprised 1 s of background stimulation (at either 5, 10, 20, 40, 60, 80 Hz) followed by 250 ms at 100 Hz burst stimulation, and was repeated twice. In some experiments slices were bath-perfused with Krebs added with 100 μM 9-Phenanthrol (Sigma-Aldrich, St. Louis, Missouri, USA).

Statistics and reproducibility: data are reported as mean ± SEM, and, unless otherwise indicated, statistical comparisons are done using paired and unpaired Student's $t$-test.

The $k$-means test was run through the Matlab "kmeans" function (see Fig. 3a). The dataset was randomly shuffled before performing the $k$-means test and the clustering was repeated 20 times with different initial centroid positions, while setting the number of cluster as an iterative parameter. Each cluster consisted of a similar range of individual observations. The "silhouette" Matlab function was used to evaluate the clustering result. The number of clusters was not assigned a priori to the dataset; rather, the algorithm converged toward an optimal solution by partitioning the dataset in separate clusters. The normality of the dataset was proved by running the Lilliefors test using the "lilliefors" Matlab function. The significance level for clustering was tested with a one-way ANOVA. The possible occurrence of batch effects was ruled out (see Supplementary Fig. 8).

Immunofluorescence: immunofluorescence of cerebellar slices was performed with a modified procedure[54,55]. Two-hundred twenty micrometer slices were fixed with freshly prepared 4% paraformaldehyde in PBS for 25 min in a Petri dish and washed in PBS. After blocking for 30 min at room temperature with 3% BSA in PBS (blocking solution), slices were incubated overnight at 4 °C with rabbit recombinant Anti-PAX6 antibody [EPR15858] (ab195045; Abcam, Cambridge, UK), diluted 1:350 in blocking solution. Slices were washed three times and then incubated for 1 h at room temperature with goat anti-rabbit IgG F(ab')2 fragment Alexa Fluor 488 conjugated (#4412; Cell Signaling technology Inc.) diluted 1:350 in blocking solution. After three washes, slices were incubated overnight at 4 °C with KO-validated rabbit Anti-TRPM4 Antibody (Cat #: ACC-044; Alomone labs, Jerusalem, Israel) diluted 1:250 in blocking solution. Slices were washed three times and incubated again for 1 h at room temperature with Donkey Fab Rabbit IgG (H&L) Antibody Rhodamine Conjugated (# 811-7002; Rockland Immunochemicals, Inc., USA) diluted 1:1000 in blocking solution. Slices were then washed three times and counterstained for 5 min at room temperature with a Hoechst solution. Finally, slices were washed and mounted in BrightMount/Plus aqueous mounting medium (ab103748; Abcam, Cambridge, UK). Slices were examined with a TCS SP5 II confocal microscopy system (LeicaMicrosystems) equipped with a DM IRBE inverted microscope (LeicaMicrosystems). Images were acquired with a 40x objective and visualized by LAS AF Lite software (Leica Microsystems Application Suite Advanced Fluorescence Lite version 2.6.0). Negative controls were performed by incubating slices with non-immune serum.

Computational modeling: the GrC model used in this study was written in Python 2.7/NEURON 7.6[58,59]. The model derived from previous ones[13,14,16] and was upgraded to account for advanced mechanisms of spike generation and conduction in the axon[15]. The ionic channels were distributed among dendrites, soma, hillock, AIS, AA, and PFs[17]. The maximum ionic conductances ($G_{max}$) were optimized using routines based on genetic algorithms (BluePyOpt)[17,60–62]. The optimization was run iteratively to improve models fitness to an experimental "template" through the automatic evaluation of "feature" values parameterizing the spike properties of the template. Out of three rounds of optimization, we obtained >600 GrC models, 150 of which were chosen randomly for further analysis. The simulation workflow was fully automated and parallelized and allowed to perform a series of protocols for: (1) parameter optimization, (2) simulation of the last generation, (3) filtering of the population based on the experimental properties, (4) simulation of each individual with protocols identical to those used experimentally.

The electrotonic structure of the GrC models was derived from[15] along with passive parameters (Supplementary Table 1 in Supplemental Material). The GrC ionic channel models and distributions were taken from previous papers and updated according to the latest literature when needed (Supplementary Table 2 in Supplemental Material). The model included Na⁺ channels (Nav1.6 with and without FHF), K⁺ channels (Kv1.1, Kv1.5, Kv2, Kv4.3, Kv3.4, Kv7), Ca²⁺ channels (Cav2.2), the newly reported TRPM4 channels coupled to Calmodulin through intracellular Ca²⁺ (ref. [63]), and a Calretinin-based Ca²⁺ buffer[64]. Synaptic transmission was modeled using the Tsodyks and Markram scheme[65] adapted as in ref. [16].

Feature extraction: GrC discharge features were extracted from experimental traces of GrCs showing regular firing. The data were taken at three current injection steps (10, 16, and 22 pA), using the "Electrophys Feature Extraction Library" (eFEL) (https://github.com/BlueBrain/eFEL)[66]. Accordingly to ref. [17], the features comprised resting membrane potential, spike width and height, fast and slow afterhyperpolarization (AHP) depth, mean spike frequency, time-to-first

spike, adaptation, and coefficient of variation of the interspike interval (ISI-CV) (Supplementary Table 3 in Supplemental Material).

Model optimization and validation: automatic optimization of maximum ionic conductances[67,68] was performed using the "Blue Brain Python Optimization Library" (BluePyOpt)[62], which is written in Python and C and uses the IBEA algorithm. Each optimization had an initial population of 288 individual, yielding a 576 final population and was performed for 12 generation, with fixed time step, at a temperature of 32 °C. A single optimization took about 5 h to be completed. After each optimization, the best individuals of the last generation were used as a guidance to reshape the ionic conductance ranges and to improve the fitness values. The parameter range for maximum ionic conductances was limited by physiological measurements as in ref. [17]. The results of optimization were validated by evaluating spike generation and conduction. A model was discarded when (1) spike generation in the AIS failed, (2) spike conduction speed or spike amplitude in the AA and PFs was decremental, (3) spike frequency was different in soma and axon (>±1 spike/s) (e.g., see ref. [15]).

Model simulations and analysis: the BluePyOpt template was customized to allow the simulation and validation of each optimized model and to improve the simulation speed by using a Python module providing access to the Message Passing Interface (MPI4py). The ionic channel conductances were uploaded from the final GrC population file. The simulations were run for 2.2 s at 32° with fixed time step[69] and with the same experimental current injections (10, 16, and 22 pA). The voltage traces were recorded and saved, for each model, from four locations: soma, AIS, final section of the AA and final section of PFs. Optimizations and simulations were carried out using 8 nodes (36 cores each) of the "HBP Blue Brain 5" cluster (BB5), located at the CSCS facility in Lugano, while coding, testing, debugging, and additional simulations were carried out on an 8core/16 threads CPU (AMD Ryzen 1800x with 32GB of ram). The simulations were analyzed by the same routines and statistical tests used for the experimental data.

**Reporting summary**. Further information on research design is available in the Nature Research Reporting Summary linked to this article.

## Data availability
The data can be requested to the authors and are available on the Knowledge Graph (Human Brain Project)[70]. https://kg.ebrains.eu/search/live/minds/core/dataset/v1.0.0/7dc5d5d5-4323-41d6-bdfd-0b841cfe7000.

## Code availability
The code is available on the Brain Simulation Platform (Human Brain Project), where models are showcased as a "live paper" on Python notebooks for optimization and simulation. https://humanbrainproject.github.io/hbp-bsp-live-papers/2019/masoli_et_al_2019/masoli_et_al_2019.html.

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

## Acknowledgements
We thank Giorgia Pellavio and Simona Tritto and the "Centro Grandi Strumenti" for assistance in histological procedures. This research was supported by the European Union's Horizon 2020 Framework Program for Research and Innovation under the Specific Grant Agreement No. 785907 (Human Brain Project SGA2) and by the MNL Project "Local Neuronal Microcircuits" of the Centro Fermi (Rome, Italy). Computing resources were provided through the EU PRACE Project TGCC 2018184373.

## Author contributions
M.T. performed electrophysiological experiments and analyzed the data; S.M. designed the models and performed the simulations; F.M. designed TRPM4 research; U.L. designed and performed immunohistochemistry; all authors contributed to paper writing and revision; E.D. coordinated research and wrote the final version of the paper.

## Competing interests
The authors declare no competing interests.
