## [Peer Review File · Communications Biology]

Reviewers' comments:

Reviewer #1 (Remarks to the Author):

This is a very interesting and beautifully illustrated paper. The immense number of granule cells at the cerebellar input stage enables a rich representation of input, possibly at a resolution surpassing that of other brain areas. It is therefore needed to characterize granule cell physiology in detail, including a search for functional subtypes, as presented here.

Major comments:

- 1) Figure 1: it is interesting that with 500ms pulses no differences in granule cell response patterns emerge, but only when using longer pulses (here 2 seconds). Differences found late in an electrophysiological response can be very informative with regard to subtype grouping, but the question arises whether such differences are physiologically relevant. This is because typical responses do not even last 500ms, certainly not 2s. The authors need to discuss this aspect of relevance in detail, including a discussion of typical durations of mossy fiber-transmitted signals.
- 2) Figure 4: the evidence for an involvement of TRPM4 is somewhat weak. The only functional test is the use of the TRPM4 blocker 9-phenanthrol. This substance is used at a concentration of 100mM (see figure legend), which is indicative of a low specificity and possible effects on other, unintended targets. The authors might want to find alternative ways to test for TRPM4 involvement, e.g. blockade by antibody binding, or the use of knockout mice. If unavailable, this issue needs to be discussed as a caveat.
- 3) Figure 4: It is obvious from the immunostainings, that TRPM4 is expressed in a vast majority of granule cells. As the paper claims that TRPM4 mediates acceleration in only one subtype of granule cells, this observation needs further discussion.

Minor comments:

- 1) Has the term 're-transmission' previously been used in this context? If not, the authors might want to consider other terms, this one does not seem to be quite accurate.
- 2) The D'Angelo group has previously reported that intrinsic plasticity takes place in granule cells. Is it possible that 'acceleration' marks cells that underwent such plasticity, and belonged to a different subtype before? In other words, are there stage-transitions?
It might be appealing to test this experimentally, but this should be left for the authors to decide as it might change the scope and focus of the paper.
- 3) P.13, last paragraph: Discussing the model, the authors argue that calmodulin binding to TRPM4 might enable acceleration. Has such interaction been shown before? There is no reference to the literature here.

Reviewer #2 (Remarks to the Author):

This manuscript challenges the view that cerebellar granule cells (GrC) are electrophysiologically and molecularly homogenous and provides ePhys and immunofluorescence evidence of GrC diversity. Further it builds on the core observation of GrC heterogeneity and provides modeling data on circuit level effects. Authors provide novel experimental evidence of GrC heterogeneity that was proposed to exist before and will lead to further interest and work.

- 1> In figure 1 to get a better sense GrC diversity authors should provide a break down of the relative

percentages of each of the electrophysiological GrCs types.

2> Authors perform k-means clustering to see structure in their IFC data in Fig3a. While k-means is probably the most well known clustering algorithm but it does require some prior conditions to be met; 1) all the clusters should have roughly the same number of observation/samples and 2) parameters should be isotropic or have similar variance, ideally normally distributed. It is not clear if those conditions were met. While accelerating population is distinct from the rest, further structure within the non-adapting, mild and strong adapting is not clear.

3> to claim that the four identified clusters are truly distinct authors should perform cross-validation by shuffling the dataset randomly, splitting the dataset into k groups, take one group as hold out and use the remaining as training set and report evaluation AUC scores

4> As control, authors should also demonstrate batch effects in their data are not driving any clustering.

5> Figure 3a, authors should explain why Kruskal-Wallis non-parametric test was used for testing significance.

6> What percentage of GrCs have TRPM4 expression in Fig 4 is not stated. It is not clear from the immunofluorescence image which subset of GrCs are positive versus negative for TRPM4. Authors should use marker like PAX6, MEIS1, TLX3 that label GrCs for proper quantification of co-localized TRPM4 protein in GrC population. The percentages of TRPM4+ GrCs should roughly be comparable to the percentages of accelerating GrCs electrophysiological types observed.

7> Fig 5 - 7 are modeling and predictions upon accounting the previously unknown TRPM4 currents. Line 7 in abstract should make that clear, specifically in the sentence where it states "Moreover, adapting GrCs were better in transmitting high-frequency mossy fiber bursts..." , that may otherwise be construed as a proven fact not a hypothesis based upon modeling data.

Reviewer #3 (Remarks to the Author):

Masoli and coworkers investigate the properties of cerebellar granule cells. Application of long (2 s) depolarizing currents elicited adapting, non-adapting, and accelerating firing properties. This finding argues against a homogeneous population of granule cells. Interestingly, the mossy fiber inputs differed in release probability depending on the firing properties of the granule cells. Furthermore, the authors argue that TRPM4 channels mediate the accelerating firing properties. Finally, modeling data indicate that the density of CaV2.2 channels critically controls the degree of the adaption.

This is a carefully-written and exciting study providing a conceptual advancement for the understanding of the function of the huge number of cerebellar granule cells. However, I have the following two technical concerns.

1. The provided TRPM4 currents are not convincing. The kinetics of the currents are very different from previous recordings (cf. ref. 21, from which the voltage protocol was used). The authors could consider artifacts like membrane ruptures, provide more convincing currents (by e.g. additional experiments or averaging of the n=4 experiments), or present and discuss the results more cautiously

(by e.g. presenting the traces of all four cells before and after application of 9-Phenanthrol).

2. The lines in Figs. 1d, 2b, 3b/c, 7b(left) look like spline interpolations or fits to experimental data. Please present the experimental data as well and describe how the lines were generated.

Minor:

The differences in frequency in Fig. 1b are not clearly explained. Please explain which two frequencies exactly were subtracted in the 0-500 and 0-2000 ms case.

Dear Reviewer #1,

We truly appreciate your insightful revision of our manuscript entitled "Parameter tuning differentiates granule cell subtypes enriching the repertoire of transmission properties at the cerebellum input stage". We carried out additional experiments and amended the manuscript accordingly. Your questions are answered here point-by-point. All the changes in the manuscript are marked in red.

Reviewer #1 (Remarks to the Author):

This is a very interesting and beautifully illustrated paper. The immense number of granule cells at the cerebellar input stage enables a rich representation of input, possibly at a resolution surpassing that of other brain areas. It is therefore needed to characterize granule cell physiology in detail, including a search for functional subtypes, as presented here.

Major comments:

1) Figure 1: it is interesting that with 500ms pulses no differences in granule cell response patterns emerge, but only when using longer pulses (here 2 seconds). Differences found late in an electrophysiological response can be very informative with regard to subtype grouping, but the question arises whether such differences are physiologically relevant. This is because typical responses do not even last 500ms, certainly not 2s. The authors need to discuss this aspect of relevance in detail, including a discussion of typical durations of mossy fiber-transmitted signals.

We thank the Reviewer for this comment. The mossy fibers are known to discharge with different patterns including long spike trains lasting for several seconds (Kase et al., 1980, *J Physiol* 300, 539-555; van Kanet al., 1993, *J Neurophysiol* 69, 74-94). Moreover, there are cellular processes triggered by neuronal activity that extend over the second timescale. It is for these reasons that previous investigations were limited and we have therefore extended the recordings time to 2secs for firing patterns and to 6 seconds for TRPM4 currents. We have now highlighted this concept in the introduction, results and discussion (Page 4, Line 11, Page 5, Line 1 and Page 22, Line 15).

2) Figure 4: the evidence for an involvement of TRPM4 is somewhat weak. The only functional test is the use of the TRPM4 blocker 9-phenanthrol. This substance is used at a concentration of 100mM (see figure legend), which is indicative of a low specificity and possible effects on other, unintended targets. The authors might want to find alternative ways to test for TRPM4 involvement, e.g. blockade by antibody binding, or the use of knockout mice. If unavailable, this issue needs to be discussed as a caveat.

We apologies for the typo in the Figure legend. As reported in the text (Page 11, Line 16), we actually used 100 μ M 9-phenanthrol. This same concentration proved efficient to inhibit TRPM4 currents in cerebellar Purkinje cells (Kim et al., 2013, *J Neurophysiol* 109: 1174–1181, 2013) and mouse prefrontal cortex neurons (Lei et al., 2014, *Front Cell Neurosci* 8:267). The reference to these papers has been added to explain our choice to use 100 μ M 9-phenanthrol. In addition, we mention the possibility to use knockout mice in future investigations (Page 22, Line 18).

3) Figure 4: It is obvious from the immunostainings, that TRPM4 is expressed in a vast majority of granule cells. As the paper claims that TRPM4 mediates acceleration in only one subtype of granule cells, this observation needs further discussion.

We thank the Reviewer for this comment and have expanded on this subject as follows (Page 22): “There are several possible reasons for why DISC current and firing acceleration were observed in just ~12% of GrCs. Apparently, immunohistochemistry showed different patterns of TRPM4 and PAX6 (related to GrC development) expression and of DAPI staining (related to chromatin folding) in GrCs (cf. Fig. 5a), which may reflect different functional states of the neurons. One can further speculate that the actual activation of TRPM4 depends on its membrane expression and physical coupling with transduction cascades involving Cav2.2, e.g. on whether a threshold sub-membrane Ca^{2+} concentration is reached upon the initial discharge next to TRPM4 channels”. We have therefore formulated an additional (but not mutually exclusive) hypothesis, in which the Ca^{2+} -dependent recruitment of TRPM4 might involve the intermediate activation of ryanodine receptors and InsP3 receptors by extracellular Ca^{2+} influx through Cav2.2.

Minor comments:

1) Has the term ‘re-transmission’ previously been used in this context? If not, the authors might want to consider other terms, this one does not seem to be quite accurate.

We thank the Reviewer for noticing. We agree that "transmission" should be used instead of "retransmission" and we have reworded the text accordingly.

2) The D'Angelo group has previously reported that intrinsic plasticity takes place in granule cells. Is it possible that ‘acceleration’ marks cells that underwent such plasticity, and belonged to a different subtype before? In other words, are there stage-transitions? It might be appealing to test this experimentally, but this should be left for the authors to decide as it might change the scope and focus of the paper.

We fully agree with the hypothesis concerning the possible involvement of GrCs intrinsic plasticity in generating different firing patterns. As the Reviewer suggests, it would be interesting to investigate this experimentally. We will surely consider the hypothesis for future experiments. For the moment, we have added a comment about plasticity in the discussion (Page 23, line 18).

3) P.13, last paragraph: Discussing the model, the authors argue that calmodulin binding to TRPM4 might enable acceleration. Has such interaction been shown before? There is no reference to the literature here.

Yes, the requirement of calmodulin for TRPM4 activation has been demonstrated (Nilius et al., 2005, J Biol Chem 280(8):6423-33). This paper clearly shows that deletion of calmodulin-binding sites at COOH-terminal impairs the Ca^{2+} -dependent activation of TRPM4 channels. The paper has been quoted in the text (Page 21, line 14).

Dear Reviewer #2,

We truly appreciate your insightful revision of our manuscript entitled "Parameter tuning differentiates granule cell subtypes enriching the repertoire of transmission properties at the cerebellum input stage". We carried out additional experiments and amended the manuscript accordingly. Your questions are answered here point-by-point. All the changes in the manuscript are marked in red.

Reviewer #2 (Remarks to the Author):

This manuscript challenges the view that cerebellar granule cells (GrC) are electrophysiologically and molecularly homogenous and provides ePhys and immunofluorescence evidence of GrC diversity. Further it builds on the core observation of GrC heterogeneity and provides modeling data on circuit level effects. Authors provide novel experimental evidence of GrC heterogeneity that was proposed to exist before and will lead to further interest and work.

1> In figure 1 to get a better sense GrC diversity authors should provide a breakdown of the relative percentages of each of the electrophysiological GrCs types.

We thank the Reviewer for noticing. Actually, in Fig. 1 we report three exemplar cells, one for each electrophysiological type. We have now added the relative percentages of the cell types in the corresponding text paragraph (Page 5, Line 5).

2> Authors perform k-means clustering to see structure in their IFC data in Fig3a. While k-means is probably the most well known clustering algorithm but it does require some prior conditions to be met; 1) all the clusters should have roughly the same number of observation/samples and 2) parameters should be isotropic or have similar variance, ideally normally distributed. It is not clear if those conditions were met. While accelerating population is distinct from the rest, further structure within the non-adapting, mild and strong adapting is not clear.

3> to claim that the four identified clusters are truly distinct authors should perform cross-validation by shuffling the dataset randomly, splitting the dataset into k groups, take one group as hold out and use the remaining as training set and report evaluation AUC scores.

4> As control, authors should also demonstrate batch effects in their data are not driving any clustering.

5> Figure 3a, authors should explain why Kruskal-Wallis non-parametric test was used for testing significance.

These four questions are related and are therefore answered together.

As the Reviewer points out, the k-means is a widely used method in cluster analysis, which minimizes the square error across clusters. K-means solves the optimization problem by assuming that each cluster has roughly equal number of observations and that all variables have similar variance. We have tested this hypothesis on the strong-adapting, mild-adapting and non-adapting GrCs. The number of clusters was not assigned a priori to the dataset; rather, the algorithm converged toward an optimal solution by partitioning the dataset in three separate clusters. The *normality* of the dataset was proved by running the Lilliefors test

on the non-adapting, mild and strong adapting populations (not shown), thus validating k-means. The significance level for these three clusters was tested with a *OneWay-ANOVA* yielding a p-value < 0.05.

The accelerating GrCs were already well separated by their own nature identified electrophysiologically and actually their inclusion into k-means resulted in a new cluster without altering the distribution of GrCs among strong-adapting, mild-adapting and non-adapting subtypes. Since the distribution of accelerating granule cells was skewed and did not meet normality, the significance level for four clusters was tested with a non-parametric Kruskal-Wallis analysis yielding a p-value < 0.01 ($p=3.7e-12$). This concept meets the practical observation that k-means is usually resistant to data normality.

We agree that the reviewer's concern is correct in principle and therefore we have preferred to adopt a conservative interpretation of k-means, which is now used only to segregate strong-adapting from mild-adapting and non-adapting GrCs subtypes, while keeping accelerating GrCs as a distinct population. This meets the conclusion that accelerating GrCs are indeed distinct because they express an additional ionic channel. In essence, clustering turns out to be the same in either cases.

In order to improve the confidence in k-means, the dataset was randomly shuffled before performing the k-means test and additional procedures were adopted in order to carry out an unsupervised data clustering. The k-means test was run using the Matlab k-means function and clustering was repeated 20 times with different initial centroid positions, while setting the number of cluster as an iterative parameter. The algorithm found four clusters as best result. To evaluate the clustering result, the silhouette Matlab function was used. The silhouette values resulted all positive and high (only 8/63 values shown a silhouette less than 0.5), thus indicating a good clusterization.

These considerations are now better explained in Methods (Page 25, line 25).

Finally, it is improbable that batch effects could have driven data clustering. Indeed, the patch-clamp experiments have been performed in different days by using different animals and newly prepared solutions. Moreover, slices were changed after each recording. Thus, even in case more GrCs were recorded per day, they belonged to different slices. We have added to Supplementary Material Fig. s7 (derived from Fig. 3a), in which a color code is used to identify cells recorded in the same day. This plot demonstrates that data were interspersed with respect to recording days, giving a convincing prove against batch effects.

6> What percentage of GrCs have TRPM4 expression in Fig 4 is not stated. It is not clear from the immunofluorescence image which subset of GrCs are positive versus negative for TRPM4. Authors should use marker like PAX6, MEIS1, TLX3 that label GrCs for proper quantification of co-localized TRPM4 protein in GrC population. The percentages of TRPM4+ GrCs should roughly be comparable to the percentages of accelerating GrCs electrophysiological types observed.

In order to address the issue, we performed new immunofluorescence experiments on the rat cerebellar granular layer by using a rabbit recombinant Anti-PAX6 antibody to label GrCs and co-staining with DAPI and a KO-validated rabbit Anti-TRPM4 antibody. As shown in Fig 5a (note that the previous panel has been moved into a separate figure), GrCs are identified by co-staining of PAX6 and DAPI. The TRPM4 expression is visible as a thin outline of the GrCs membrane, while PAX6 and DAPI fluorescence is confined to the nuclear region. Clearly, this new immunolabeling helps to better define the localization of TRPM4 in GrCs and we thank the reviewer for suggesting. Visualization is clearly much better now. Nonetheless, we

believe that correlating the number of GrCs with TRPM4 staining with that of GrCs with firing acceleration is unpractical. Channel expression is a *sine qua non* but then the channel needs to be coupled to calcium and the calcium system must be activated for accelerating firing. And this cannot be assessed in the present measurements. We agree this is an outstanding issue but unfortunately it cannot be answered at the moment and would merit a future investigation. We have explained all of this in the discussion and formulated two not mutually exclusive hypotheses to explain why, even though TRPM4 is largely expressed in PAX6+ GrCs, firing acceleration arises only in just ~12% of GrCs (Page 22).

7> Fig 5 - 7 are modeling and predictions upon accounting the previously unknown TRPM4 currents. Line 7 in abstract should make that clear, specifically in the sentence where it states “Moreover, adapting GrCs were better in transmitting high-frequency mossy fiber bursts...” , that may otherwise be construed as a proven fact not a hypothesis based upon modeling data.

Thank you for noticing, the abstract has been corrected.

Dear Reviewer #3,

We truly appreciate your insightful revision of our manuscript entitled "Parameter tuning differentiates granule cell subtypes enriching the repertoire of transmission properties at the cerebellum input stage". We carried out additional experiments and amended the manuscript accordingly. Your questions are answered here point-by-point. All the changes in the manuscript are marked in red.

Reviewer #3 (Remarks to the Author):

Masoli and coworkers investigate the properties of cerebellar granule cells. Application of long (2 s) depolarizing currents elicited adapting, non-adapting, and accelerating firing properties. This finding argues against a homogeneous population of granule cells. Interestingly, the mossy fiber inputs differed in release probability depending on the firing properties of the granule cells. Furthermore, the authors argue that TRPM4 channels mediate the accelerating firing properties. Finally, modeling data indicate that the density of CaV2.2 channels critically controls the degree of the adaption.

This is a carefully-written and exciting study providing a conceptual advancement for the understanding of the function of the huge number of cerebellar granule cells. However, I have the following two technical concerns.

1. The provided TRPM4 currents are not convincing. The kinetics of the currents are very different from previous recordings (cf. ref. 21, from which the voltage protocol was used). The authors could consider artifacts like membrane ruptures, provide more convincing currents (by e.g. additional experiments or averaging of the n=4 experiments), or present and discuss the results more cautiously (by e.g. presenting the traces of all four cells before and after application of 9-Phenanthrol).

We thank the Reviewer for this comment. Of course, the fast TRPM4 currents described in the present investigation are not due to artifacts, as the Giga-seal and the series resistance were continuously monitored during recordings. The unstable cells were discarded during the analysis. The Referee is right when noticing that, unlike cerebellar Purkinje cells, GrCs display a burst of DISCs after the depolarization protocol. We carried out more recordings to address this issue and confirmed that the same voltage-clamp protocol adopted in Purkinje cells induced a burst-like pattern of DISCs in accelerating GrCs, while it was ineffective in the non-accelerating ones. In Fig. 4c we now display average current traces for both accelerating and strong-adapting GrCs. The average current taken from the whole set of accelerating GrCs is rather smooth and show kinetics compatible with the currents observed in pancreatic β -cells, as quoted in the text. We discussed in the text (Page 22) that, rather than mirroring CaV2.2 activation (as described in Purkinje cells), the delayed Ca²⁺-dependent recruitment of TRPM4 might involve the intermediate activation of ryanodine receptors and InsP3 receptors by extracellular Ca²⁺ influx through Cav2.2

2. The lines in Figs. 1d, 2b, 3b/c, 7b(left) look like spline interpolations or fits to experimental data. Please present the experimental data as well and describe how the lines were generated.

We have modified the plot of Figs. 1d, 2b, 3b/c, 7b(left) superimposing data points over the spline interpolations.

Minor:

The differences in frequency in Fig. 1b are not clearly explained. Please explain which two frequencies exactly were subtracted in the 0-500 and 0-2000 ms case.

The histograms of Fig. 1b show the average frequencies calculated in the initial and final 500 ms time-windows of current injection (0-500 ms and 1500-2000 ms, respectively). Within each of these time windows, the frequency was measured at the beginning and the end by averaging 5 interspike intervals.

REVIEWERS' COMMENTS:

Reviewer #1 (Remarks to the Author):

All my previous concerns have been addressed.

Reviewer #2 (Remarks to the Author):

Authors have addressed my concerns and hence I believe that this manuscript in its current form should be accepted for publication.

Reviewer #3 (Remarks to the Author):

The authors have convincingly addressed my concerns. I support publication but have one more suggestion.

I understand that "we have modified the plot of Figs. 1d, 2b, 3b/c, 7b(left) superimposing data points over the spline interpolations." but I think the data points should have error bars (e.g. standard error of the mean).